# Biochemical and molecular characterization of the SBiP1 chaperone from *Symbiodinium microadriaticum* CassKB8 and light parameters that modulate its phosphorylation

**Raúl Eduardo Castillo-Medina, Tania Islas-Flores, Estefanía Morales-Ruiz, Marco A. Villanueva** [ID] *

Unidad Académica de Sistemas Arrecifales, Instituto de Ciencias del Mar y Limnología, Universidad Nacional Autónoma de México-UNAM, Puerto Morelos, Quintana Roo, México

* marco@cmarl.unam.mx

## Abstract

The coding and promoter region sequences from the BiP-like protein SBiP1 from *Symbiodinium microadriaticum* CassKB8 were obtained by PCR, sequenced and compared with annotated sequences. The nucleotides corresponding to the full sequence were correctly annotated and the main SBiP1 features determined at the nucleotide and amino acid level. The translated protein was organized into the typical domains of the BiP/HSP70 family including a signal peptide, a substrate- and a nucleotide-binding domain, and an ER localization sequence. Conserved motifs included a highly conserved Thr513 phosphorylation site and two ADP-ribosylation sites from eukaryotic BiP's. Molecular modeling showed the corresponding domain regions and main exposed post-translational target sites in its three-dimensional structure, which also closely matched *Homo sapiens* BiP further indicating that it indeed corresponds to a BiP/HSP70 family protein. The gene promoter region showed at least eight light regulation-related sequences consistent with the molecule being highly phosphorylated in Thr under dark conditions and dephosphorylated upon light stimuli. We tested light parameter variations that could modulate the light mediated phosphorylation effect and found that SBiP1 Thr dephosphorylation was only significantly detected after 15–30 min light stimulation. Such light-induced dephosphorylation was observed even when dichlorophenyl dimethyl urea, a photosynthesis inhibitor, was also present in the cells during the light stimulation. Dephosphorylation occurred indistinctly under red, yellow, blue or the full visible light spectra. In additon, it was observed at a light intensity of as low as 1 μmole photon $m^{-2}$ $s^{-1}$. Our results indicate that: a) SBiP1 is a chaperone belonging to the BiP/HSP70 family proteins; b) its light-modulated phosphorylation/dephosphorylation most likely functions as an activity switch for the chaperone; c) this light-induced modulation occurs relatively slow but is highly sensitive to the full spectrum of visible light; and d) the light induced Thr dephosphorylation is independent of photosynthetic activity in these cells.

**Data Availability Statement:** All relevant data are within the paper and its Supporting Information files.

**Funding:** This work was supported by grant 285802 from the National Council of Humanities Science and Technology of Mexico (CONAHCyT), awarded to M.A.V.; R.C.-M and E.M.-R. were supported by PhD (255464) and post-doctoral fellowships, respectively, also from CONAHCyT. The funders had no role in study design, data collection and analysis, decision to publish, or preparation of the manuscript.

**Competing interests:** The authors have declared that no competing interests exist.

# Introduction

Photosynthetic dinoflagellates of the Symbiodiniaceae family are marine microorganisms that live either freely in the seas or captured within a mutualistic symbiotic relationship in the tissues of some marine invertebrates [1]. *Symbiodinium microadriaticum* CassKB8 dinoflagellates (hereinafter referred to as CassKB8) belong to this family and are usually found in symbiosis with the jellyfish *Cassiopea xamachana*. The ability of these microorganisms to live freely in the water column makes them amenable for *in vitro* culture. During their physiological stages in either condition, they are subjected to the normal day/night photoperiod cycles, which means that they must contend with two transitional periods where internal cellular metabolism drastically changes, i.e., from photosynthesis to photorespiration and viceversa. Thus, they require fine sensing mechanisms to respond to changing light conditions and other stimuli from the environment that trigger signal-transduction pathways. Such signaling cascades must function with specific sets of receptor, adapter and effector proteins [2, 3]. For example, the CassKB8 RACK1 homolog SmicRACK1 was shown to follow an expression pattern that closely paralleled the peaks of each light/dark cycle, and expression decreased at the end of each cycle [4]. Those gene expression changes could be further fine-tuned via post-translational modifications since the RACK1 homologs themselves can undergo phosphorylation/dephosphorylation [5–7].

One particular and fundamental switch to turn on and off signaling within a cell is through such exquisitely regulated phosphorylation and dephosphorylation reactions via protein kinases and phosphatases, respectively, on key target molecules. Protein kinases and phosphatases from eukaryotic organisms belong to superfamilies formed by numerous copies [8–10]. In Symbiodiniaceae, the kinase domains are the second most abundant domains in the so far sequenced genomes, with 374 such domains reported from *Fugacium kawagutii* to 756 from *Breviolum minutum*, and 869 gene families (Pfam PF00069) showing a potential for abundant kinase activities [11, 12]. For example, 177 transcripts from more than 20 serine/threonine-protein kinase families significantly changed their expression levels when *Symbiodinium* sp. (clade F, ITS2) was thermally stressed [13]. Conversely, fewer information is available on protein phosphatases from Symbiodiniaceae although more than 250 phosphatase-related nucleotide sequences are reported on the NCBI database.

Many other proteins are also regulated via phosphorylation including BiP chaperones [14–17]. These BiP chaperones belong to the HSP70 protein family from the endoplasmic reticulum (ER), where they assist the folding and assembly of newly synthesized proteins when they are translocated into the ER, and they also associate with misfolded and/or underglycosylated proteins [18, 19]. They are both transcriptionally and post-translationally regulated through AMPylation, ADP-ribosylation, methylation, ubiquitination, and acetylation. Additionally, it has been reported that significant BiP activity control in eukaryotes occurs via phosphorylation/dephosphorylation [16].

Phosphorylation/dephosphorylation of proteins upon light stimuli have only recently been reported in three photosynthetic dinoflagellate species including CassKB8 [20]. Among several proteins whose phosphorylation levels responded to a light stimulus, SBiP1, an HSP70- and BiP-like 75 kDa protein was identified in all species analyzed with a high Thr phosphorylation level during 12 h of sustained darkness but decreased after a 30 min light stimulus in all three species. SBiP1 is present as several differentially phosphorylated isoforms, indicating a likely fine regulation and differential function upon the light stimulus after the dark/light transition phase [17, 20]. These findings were novel as no previous reports existed documenting that BiP phosphorylations could be modulated by light. Furthermore, we showed that heat stress, either short-term induced or sustained, promoted the dephosphorylation of the chaperone under prolonged darkness [17], which correlates with its activation [15, 16].

In order to obtain further knowledge on the SBiP1 molecule and on the parameters during the perception of the stimulus that modulates this protein by phosphorylation, we carried out a more detailed analysis of the sequence and molecular features of the protein and determined the light spectral and intensity regimes that triggered the response. We found key features, motifs and sequences of the molecule that place it as a typical protein of the BiP/HSP70 family and confirmed its presence in all reported sequences from Symbiodiniaceae. In addition, we determined that the light-induced dephosphorylation of SBiP1 occurred regardless of the wavelength and intensity of the light stimulus, and that the light-stimulated dephosphorylation response was independent of the onset of photosynthesis in CassKB8.

## Materials and methods

### CassKB8 cell cultures

Dinoflagellate cultures of CassKB8 (also known as MAC-CassKB8 and previously classified as clade A *Symbiodinium*) originally isolated from the jellyfish *Cassiopea xamachana*, were a kind gift of Dr. Mary Alice Coffroth (State University of New York at Buffalo). Cultures were routinely maintained in our laboratory in ASP-8A medium under photoperiod cycles of 12 h light/dark at 26°C. Light intensity for routine culture was maintained at 80–120 μmole photon $m^{-2} s^{-1}$.

### Antibodies and reagents

The polyclonal anti-phosphothreonine (anti-pThr) antibodies were from Cell Signaling Technology™, Inc. (Danvers, MA; cat. 9381S) and Abcam (Cambridge, UK; cat. Ab9337). For loading controls and normalization in quantitation analyses either, a monoclonal anti-actin antibody (Cat. No. N350, originally purchased from Amersham) known to cross-react with actin from many species [21] including Symbiodiniaceae [20, 22], or the purified IgG fraction (~ 1 mg/ml IgG as stock) of a rabbit anti-SBiP1 polyclonal antibody [17], were used. Alkaline-phosphatase (AP) conjugated polyclonal anti-rabbit IgG and anti-mouse IgG antibodies raised in goat were from Zymed®-Life Technologies (Grand Island, NY). Reagents 5-bromo-4-chloro-3-indolyl phosphate (BCIP) and nitro blue tetrazolium (NBT) were from Promega (Madison, WI). The inhibitor of photosystem II (PSII) 3-(3,4- dichlorophenyl)-1,1-dimethy-lurea (DCMU) was from Sigma. All other reagents were from Sigma.

### Polymerase chain reaction

Preparation of cDNA was carried out as described previously [23]. RNA was extracted with Tri-Reagent (Sigma) according to the manufacturer's protocol. The specific Dino-SL (S1 Table; [23]) and SBiPRv (S1 Table) oligonucleotides were used to amplify the *SBiP1* ORF by PCR as follows: CassKB8 cDNA was mixed with 0.2 μM of the respective oligonucleotides, 1X Dream Taq Buffer, 0.2 mM dNTP mix and 1.25 U of Dream Taq DNA Polymerase (Thermo-Fisher, Waltham, MA) in a final volume of 50 μl. The PCR reaction settings were as follows: step 1, 95°C for 3 min; step 2, 95°C for 30 s; step 3, 60°C for 30 s; and step 4, 72°C for 1.5 min. Steps 2–4 were repeated 30 times. Next, step 5, 72°C for 10 min; and step 6, 12°C for 15 min. Genomic DNA from CassKB8 was extracted as follows: cells were lysed in extraction buffer (4.06 M GuSCN, 0.1 M Tris-HCl, pH 6.4, 0.2 M EDTA, pH 8.0 and 1% TX-100) at 56°C for 1 h. The lysate was centrifuged at 12,000 x *g* and the supernatant was recovered and transferred to silica columns followed by centrifugation at 12,000 x *g* for 30 s. The unbound material was discarded, the column added with 500 μl binding buffer (4.06 M GuSCN, 0.1 M Tris-HCl pH 6.4, 0.2 M EDTA pH 8), and centrifuged at 12,000 x *g* for 30 s (this step was repeated until the

filtrate was clear). Then, the column was washed 3 times with 600 µl washing buffer (70% etanol, 10 mM NaCl) and eluted with nuclease-free water which was used as template. For PCR amplification of the *SBiP1* gene containing its promoter (3,481 bp's), CassKB8 genomic DNA was mixed with 0.2 µM of the SmicBiP PromF and hsp75STOP_BH1 Rv oligonucleotides (S1 Table), 1X Dream Taq Buffer, 0.2 mM dNTP mix and 1.25 U of Dream Taq DNA Polymerase (ThermoFisher). The PCR reaction settings were: step 1, 95˚C for 3 min; step 2, 95˚C for 30 s; step 3, 64˚C for 30 s; and step 4, 72˚C for 3.5 min. Steps 2–4 were repeated 35 times. Next, step 5, 72˚C for 5 min; and step 6, 12˚C for 5 min. The PCR product spanning the ORF was cloned into the pGEM™-T Easy vector, and this construction was sent for sequencing using oligonucleotides that cover the full length of the ORF (S1 Table). The genomic PCR amplicon was sequenced using oligonucleotides that span the sequence from nucleotides -815 to 2,665 (S1 Table). Sequencing was carried out at the facility of the Institute of Biotechnology-UNAM.

## SBiP1 sequence analyses

The *SBiP1* sequences were compared against the reported *S. microadriaticum* [24] and CassKB8 [25] sequences to dissect the correct start and termination codons. Promotor and regulatory regions were analyzed with the PLACE program [26]. The correctly placed ORF was translated to aminoacids and subjected to prediction analyses for identification of the signatures with: SignalP 5.0 [27] for signal peptide and cleavage sites (based on a combination of several artificial neural networks and hidden Markov models to predict cleavage sites and a signal peptide/non-signal peptide); InterProScan [28] for the nucleotide and substrate binding domains, the linker sequence, and the C-terminal subdomain; and the Prosite database [29] and NetPhos 3.1 [30] for phosphorylation target sites. α- and β-substrate binding subdomains were assigned based on the BiP sequences reported by [31]. The HSP70 family signatures and the Endoplasmic Reticulum targeting sequence His-Asp-Glu-Leu were identified with Motif Scan [32]. Finally, ADP-ribosylation sites were determined by feeding the sequence into the ADPredict online platform according to [33]. For the three-dimensional structure analysis, the primary amino acid sequence of SBiP1 was submitted to AlphaFold2 using the AlphaFold2 ColabFold notebook v1.5 [34]. The PDB file was then visualized and colored with UCSF ChimeraX, developed by the Resource for Biocomputing, Visualization, and Informatics at the University of California, San Francisco [35–37].

## Protein electrophoresis in SDS-PAGE gels and western blot

The protein extracts were separated in discontinuous denaturing gels [38], of 10% polyacrylamide in the separation zone [375 mM Tris-HCl, pH 8.8; 10% acrylamide/bis-acrylamide (29:1); 0.1% SDS; 0.1% ammonium persulphate (APS); 0.106% N, N, N, N'-tetramethylethylenediamine (TEMED)], and 4% polyacrylamide in the stacking zone [125 mM Tris-HCl, pH 6.8; 4% acrylamide/bisacrylamide (29:1); 0.1% SDS; 0.1% APS; 0.066% TEMED], in a Mini-PROTEAN™ 3 System (Bio-Rad, Hercules, CA). After electrophoresis, the proteins were transferred to PVDF membranes in "friendly buffer" (25 mM Tris-HCl, 192 mM glycine, 10% isopropanol; [39]) at a constant current of 300 mA for 1 h. The membranes were blocked in a solution of 3% bovine serum albumin (BSA) in PBS (2.79 mM $NaH_2PO_4$, 7.197 mM $Na_2HPO_4$, 136.9 mM NaCl, pH 7.5) added with 0.01% TX-100 (PBS-T) for 1 h at 50˚C, with gentle agitation. After blocking, the primary antibodies, anti-pThr (Cell Signaling, 1:2,500), anti-actin (1:1,000), or anti-SBiP1 (0.404 µg/ml IgG fraction) in PBS-T, were added to the membranes and incubated overnight with gentle rocking at 25˚C. Alternatively, when Abcam anti-pThr antibodies were used, TBS-T (20 mM Tris-Base, 150 mM NaCl, 0.01% TX-100, pH 7.5) was the buffer for the BSA blocking solution (5% BSA) and primary antibody dilution

(1:500). Blocking for at least 2 h and all other incubations were carried out at 4˚C. After over-night incubation, the membranes were washed five times, 5 min each, in PBS-T or TBS-T, and incubated with the appropriate secondary antibodies (alkaline-phosphatase conjugated anti-rabbit IgG or anti-mouse IgG) at 1:2,500 dilution in PBS-T or TBS-T for 2 h at 25 or 4˚C, buffer and temperature depending on the primary antibody used. Subsequently, the membranes were washed again five times, 5 min, and the final wash was followed by a brief rinse with alkaline developing solution (100 mM Tris-HCl, 150 mM NaCl, 1 mM $MgCl_2$, pH 9) at 25˚C. Finally, the membranes were developed with a commercial solution of 5-bromo-4-chloro-3-indolyl phosphate (BCIP) and nitro blue tetrazolium (NBT) according to the manufacturer (Promega, Madison, WI), in alkaline developing solution. In the case of anti-pThr from Cell Signaling, development was carried out in PBS-T to ensure more astringency and less background. It is important to note that we have previously determined that incubations in PBS-T or TBS-T yielded identical results when anti-pThr antibodies from Cell Signaling were used [20].

## Thr phosphorylation analysis of SBiP1 from CassKB8 cells adapted to darkness, after different illumination times, intensities, and spectra

Six-d-old cultures from CassKB8 from three biological replicates collected 2 h before the diurnal phase of the photoperiod, were concentrated by centrifugation at 2,600 x $g$ for 3 min and suspended in fresh ASP-8A medium to reach 8–14 x $10^5$ cells/ml. The cells were placed in equal aliquots into 15 ml Falcon tubes wrapped with aluminum foil and incubated for 12 h under darkness at 26˚C, then each tube was exposed to the following: a) a light intensity of ~ 100 µmole photon $m^{-2}$ $s^{-1}$ for 30 s, 1, 5, 15, and 30 min; b) light intensities of 1, 10, 40, and 650 µmole photon $m^{-2}$ $s^{-1}$; or c) blue (450–490 nm), yellow (490–750 nm), red (610–750 nm), or white (full spectrum) lights at ~ 100 µmole photon $m^{-2}$ $s^{-1}$. The different light intensities were achieved by shading a LED lamp with various layers of shade mesh and were measured using a submersible spherical micro quantum sensor (US-SQS, Walz, Germany) connected to a LI-COR 1400 light meter (Li-Cor, USA). After the light stimulation, the cells in the tubes were sedimented by centrifugation at 2,600 x $g$ for 1 min at 26˚C and immediately suspended in 300 µl Laemmli buffer [38], supplemented with 0.2 mM $Na_3O_4V$, 10 mM NaPPi and a cocktail of protease inhibitors (Complete™, Roche, Basel, Switzerland), mixed with ~ 250 µl total volume of glass beads (465–600 µm diameter; Sigma-Aldrich) and lysed with a MINI-BEAD BEATER™ (Biospec products). The lysate was heated at 95˚C for 5 min, centrifuged at 12,000 x $g$ for 10 min, and the supernatant used for analysis. Equal loads of proteins were adjusted with equal aliquots of cells and standardized additionally by staining with coomassie blue in 10% polyacrylamide SDS-PAGE gels. Finally, the extracts were analyzed by western blot with either anti-pThr (Cell Signaling), or anti-actin (for light intensities and spectra assays) or anti-SBiP1 (for all other assays) as reference for loading control and normalization. The bands from the triplicate samples were captured with a ChemiDoc-It 2™ Imager (UVP-Analytik Jena, Upland, CA, USA), analyzed by densitometry with VisionWorks LS software, and normalized with the band intensities from the internal actin or SBiP1 controls. The results were integrated into graphs displaying the average band intensity of the three biological replicates for each treatment.

## Thr phosphorylation analysis of SBiP1 from CassKB8 cells in the presence of DCMU during the transition of darkness to light

Six d-old CassKB8 cultures were sedimented by centrifugation at 2,600 x $g$ for 3 min and suspended in fresh ASP-8A medium to reach a concentration of 4–6 x $10^5$ cells/ ml. Three equal

portions of 40 ml in Erlenmeyer flasks wrapped with aluminum foil were incubated for 12 h at 26˚C under darkness (nocturnal phase). The cultures were then treated 2 h (after 10 h of darkness) before the change to light with 0.01 mM 3-(3,4- dichlorophenyl)-1,1-dimethylurea; (DCMU; 400 μl of a 1 mM stock), or 400 μl vehicle (EtOH) as negative control. Protein extraction was carried out at either 12 h darkness, 30, or 60 min of light for each treatment, as described above. The proteins were analyzed by 10% SDS-PAGE gels, and western blot with either anti-pThr (Abcam), or anti-SBiP1 antibodies for loading control and normalization. The bands on the blots were analyzed by densitometry and normalized as above. The data from three independent experiments were averaged, subjected to ANOVA analysis, and plotted.

## Assessment of the photochemical efficiency of CassKB8 cells treated with DCMU

In parallel with the cell treatments with DCMU, the photochemical efficiency was determined using a pulse amplitude modulated fluorometer (diving PAM, Waltz, Germany) at 26˚C (n = 3). Two ml of cell suspension containing approximately $1 \times 10^6$ cells/ml were placed in a spectrophotometer cell and constantly stirred. Measurements were made after 2 h of DCMU treatment in the dark, and every 10 min for 1 h under light.

## Results

### Correct annotation of SBiP1, sequence analysis and detection in annotated Symbiodiniaceae genomes and transcriptomes

PCR using specific SBiP1 oligonucleotides followed by sequencing of the amplicon revealed the full ORF (S1 Fig). This ORF sequence was slightly different from the previously reported SBiP1 sequence with a length of 689 amino acids [20]. Therefore, we dissected the correct sequence of SBiP1 to 1,965 nucleotides and 655 amino acids (Fig 1B; S1 Fig). We submitted the correct sequence to the GenBank with accession number OP429595. The translated amino acid sequence consisted of clear domains and signatures of the BiP/HSP70 family that included an N-terminal signal peptide of 18 amino acids (Fig 1A, SP; Fig 1B, 1-18; S2 Fig), a 384 amino acid long nucleotide-binding domain (Fig 1A, NBD; Fig 1B, 22–399), a 13 amino acid linker sequence (Fig 1A, LK; Fig 1B, 403–415), and a substrate-binding domain consisting of two subunits: a 111 amino acid beta subunit (Fig 1A, SBDβ; Fig 1B, 416–526), and a 125 amino acid alpha subunit (Fig 1A, SBDα; Fig 1B, 527–651). In addition, at the C-terminus, an 86 amino acid long C-terminal subdomain typical of the HSP70 family proteins (Fig 1A, C-ter; Fig 1B, 555–640), as well as the ER-localization sequence His-Asp-Glu-Leu (HDEL) (Fig 1B, 652–655), were observed. Three typical BiP signatures were detected in stretches of amino acids 32–39, 218–231, and 355–369 (Fig 1B, grey highlight). Several amino acid sites with a high score as targets for phosphorylation were detected for both Thr (Fig 1B, green highlight) and Ser (Fig 1B, yellow highlight). Among these, two phosphorylation sites were also present on the sequence; one at Tyr 309 (Fig 1B, magenta highlight), and a highly conserved Thr 513 (Fig 1B, green highlight with asterisk; S3 Fig), which has also been experimentally shown to become phosphorylated in BiP from *Chlamydomonas reinhardtii* [15]. Another important feature was the presence of the two conserved Arg residues at positions 465 and 487 (Fig 1B, framed in purple; S3 Fig) important for ADP-ribosylation [16, 40]. The 3D model from the amino acid sequence obtained by the AlphaFold2 ColabFold notebook, shows the different conserved motifs (Fig 2A). The NetPhos 3.1 predicted phosphorylation sites for Thr with a score above 0.5 and surface exposed were localized on the 3D structure (Fig 2B, black); of

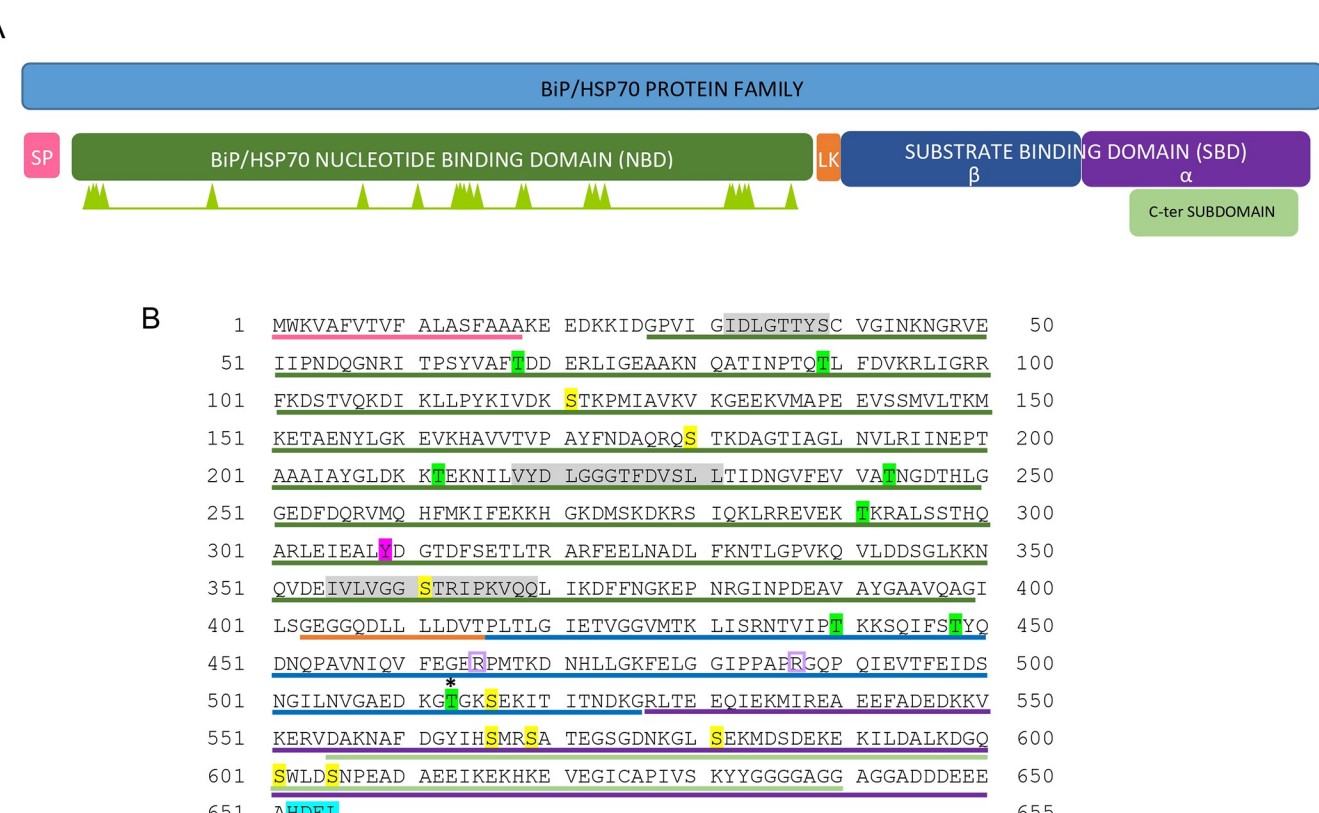

**Fig 1. Domain analysis and amino acid translation of the corrected SBiP1 sequence.** A. The typical domains of a BiP/HSP70 family protein were detected. A signal peptide (SP), nucleotide-binding (NBD), linker sequence (LK), substrate-binding (SBD), and a C-terminal (C-ter) domains are present. B. The amino acid sequence shows highly conserved and likely phosphorylatable Thr (green highlight) and Ser (yellow highlight) residues, from which the residue at position 513 (asterisk) has been experimentally shown to undergo phosphorylation [15]. Two highly conserved and likely ADP-ribosylatable Arg residues [16] at positions 465 and 487 (framed in purple) were also present. The ER retention signal His-Asp-Glu-Leu (HDEL) is also observed (blue highlight). The colors on the domains in (A) correspond to the colors of the underlined aminoacids in (B). The sequence was annotated in the GenBank with accession number OP429595.

particular interest was the observation that the conserved, experimentally demonstrated Thr513 [15] was also surface exposed (Fig 2B, yellow, framed in dotted square). In addition, the conserved Arg 465 and 487 reported as targets of ADP-ribosylation [16, 40] were observed exposed on the surface of the 3D-structure (Fig 2A, pale green in SBDβ). When the 3D structure of SBiP1 was superimposed with that of *H. sapiens* BiP, both were observed to closely match, confirming a conserved structural homology (S4 Fig). The nucleotide sequence and intron/exon distribution of the reported CassKB8 *SBiP1* gene [25] are shown in S5 Fig. The gene encompassed by 2,665 nt's contained 7 introns (located at nt's 105–228, 604–688, 791–889, 1,309–1,365, 1,510–1,593, 1,713–1,824, 2,067–2,177), and 8 exons (at nt's 1–104, 229–603, 689–790, 890–1,308, 1,366–1,509, 1,594–1,712, 1,825–2,066, 2,178–2,665); this sequence transcribed into an ORF of 1,968 nt's (Fig 1B; S5 Fig). Within the sequence, a substitution of A for C (S5 Fig; A130C, His44Asn, shaded in orange) that changed His 44 for Asn, and a synonymous substitution of A for G (S5 Fig; A780G, Gln260Gln, shaded in yellow), were observed. This *SBiP1* promoter sequence obtained by genomic PCR was identical to the equivalent region in the reported CassKB8 *SBiP1* gene sequence [25]. The search for *cis*-acting regulatory elements in this sequence (from -1 to -700 nt) rendered 17 elements, of which 12 were different (Fig 3A). Among such 12 *cis*-acting regulatory elements, 9 corresponded to light regulated

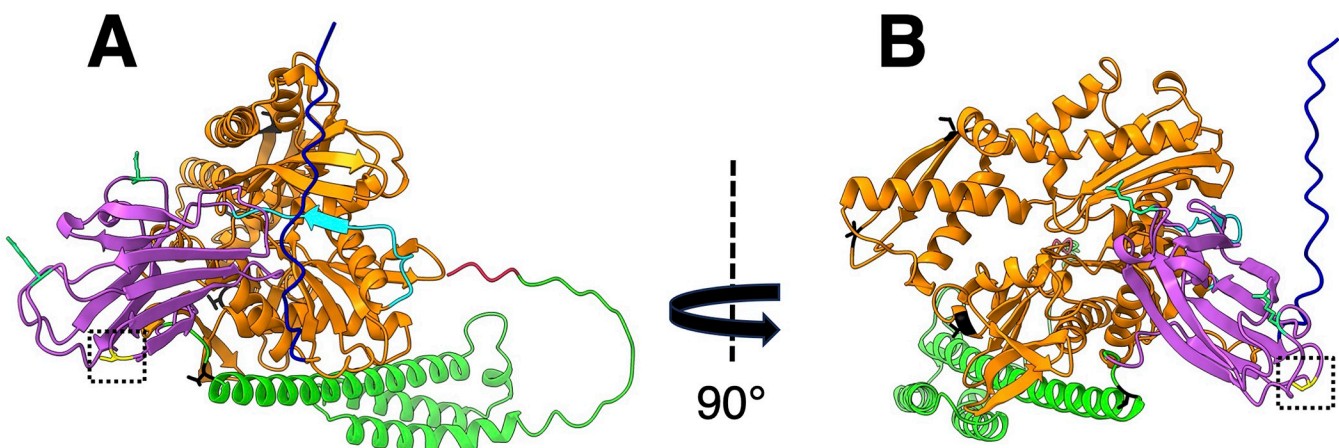

**Fig 2. Structural aspects of SBiP1 after assembly of the sequence with AlphaFold2.** In blue, the signal peptide; in orange, the nucleotide binding domain (NBD); in cyan, the hydrophobic linker; in green, the alpha subunit of the substrate binding domain (SBDα); in fuchsia, the beta subunit of the SBD (SBDβ); and in red, the ER-retention signal. Finally, surface exposed and high probability Thr-phosphorylation and ADP-ribosylation residues are shown in black and pale green, respectively. The conserved Thr 513 experimentally demonstrated to undergo phosphorylation [15] is shown in yellow (framed in dotted square) and is also observed exposed on the molecule, readily available to modifying enzymes.

elements, one related to heat shock protein genes, other related to $Ca^{+2}$-responsive upregulated genes, and the last one was identified as a copper-response element also involved in oxygen-response (Fig 3A and 3B).

Finally, we had previously renamed the protein from SmicHSP75 to SBiP1 due to its detection in several species of Symbiodiniaceae [17]. Consequently, we searched the homologous sequences in all annotated Symbiodiniaceae genomes and transcriptomes to date by BLAST [41] analysis against the correct SBiP1 sequence and identified the presence of SBiP1 homologs in all cases (Table 1).

## Thr dephosphorylation of SBiP1 from CassKB8 cells adapted to darkness occurs after 15–30 min of light stimulation even at low intensity and under various light spectra

CassKB8 cells adapted to darkness showed a high level of Thr phosphorylation (Fig 4A and 4B, lane SBiP1-p and black bar, 12 h Dark, respectively), as previously reported [17, 20]. Since we had previously tested relatively long times of light exposure (30–240 min), we shortened the light exposure range of the dark-adapted cells after the onset of light. When they were stimulated with light at ~ 100 μmole photon m$^{-2}$ s$^{-1}$ for shorter time intervals (0.5, 1, 5, 15, and 30 min), we observed that SBiP1 dephosphorylation on Thr was visible after 15 min of light stimulation (Fig 4A and 4B, lane SBiP1-p and white bar, 15 min, respectively), and was still observed at apparently higher levels after 30 min of light (Fig 4A and 4B, lane SBiP1-p and white bar, 30 min, respectively). The observed dephosphorylation effect was shown to be statistically significant when P-Thr from SBiP1 (Fig 4A, lanes SBiP1-p) was quantitated and normalized to the total SBiP1 (Fig 4A, lanes SBiP1) from three biological replicates (Fig 4B, asterisks on white bars). Therefore, since we have previously determined that the level of Thr SBiP1 phosphorylation did not change significantly after 30–240 min of light exposure [17, 20], a 30 min light stimulation was used for all subsequent assays. When dark-adapted cells were stimulated for 30 min at the same light intensity with wavelengths corresponding to blue, yellow, or red light spectra (Fig 5C), the light induced SBiP1 dephosphorylation was observed (Fig 5A, SBiP1-p lanes B, Y, R; Fig 5B, colored bars) with respect to the dark unstimulated cells

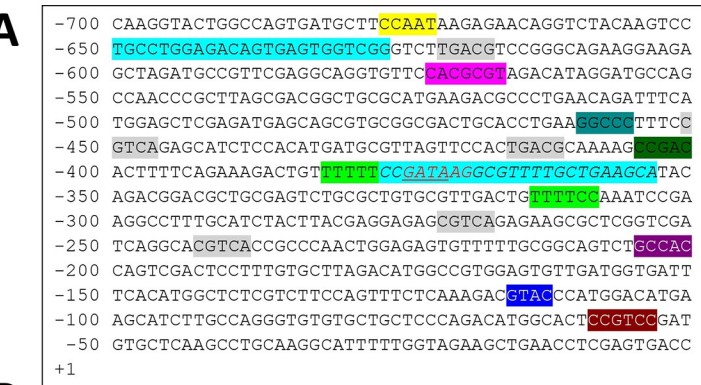

**Fig 3. Promoter *cis*-acting regulatory element analysis of the *SBiP1* gene.** The promoter region from the *SBiP1* gene was analyzed to find the regulatory *cis*-acting elements with the PLACE (plant *cis*-acting regulatory DNA elements; [26]. A. The sequence showing the 700 nucleotides upstream of the *SBiP1* gene that were used to carry out the analysis. B. The distinct regulatory elements found within the sequence showing their position (nt), name, sense, sequence, and corresponding description.

(Fig 5A, SBiP1-p lane Dark, 12 h; Fig 5B, black bar), and such effect was similar to the one induced by the white light control (Fig 5A, SBiP1-p lane W; Fig 5B, white bar). The dephosphorylation effect was statistically significant (Fig 5B, asterisks on bars) from three biological replicates when actin (Fig 5A, lanes "actin") was used as internal loading and normalization control. Interestingly, when various light intensities were used for the stimulation, Thr dephosphorylation of SBiP1 was observed at the lowest intensity of 1 μmole photon m$^{-2}$ s$^{-1}$ (Fig 6A and 6B, SBiP1-p lane 1 light and dark grey bar 1, respectively), but it occurred in an intensity-dependent fashion with the highest dephosphorylation observed at 40 μmole photon m$^{-2}$ s$^{-1}$ (Fig 6A and 6B, SBiP1-p lane 40 light and light grey bar 40, respectively); however, a higher light intensity of 650 μmole photon m$^{-2}$ s$^{-1}$, did not produce a further SBiP1 dephosphorylation (Fig 6A and 6B, SBiP1-p lane 650 light and white bar 650, respectively). These results indicated that SBiP1 dephosphorylation on Thr in CassKB8 cells occurs at very low light intensity and regardless of the visible light wavelength, but it requires at least 30 min of light stimulus and 40 μmole photon m$^{-2}$ s$^{-1}$ to reach its maximum.

**Table 1. Accession numbers and identities of BiP sequences from Symbiodiniaceae detected in the GenBank.**

| Organism | Name | Query cover | Percent identity | Accession |
|---|---|---|---|---|
| **PROTEIN** | | | | |
| *Symbiodinium sp. CCMP2592* | BIP5 | 96% | 97.79 | CAE7227403.1 |
| *Symbiodinium microadriaticum* | BIP5 | 96% | 96.68 | OLP91134.1 |
| *Symbiodinium pilosum* | BIP5 | 96% | 79.03 | CAE7221339.1 |
| *Symbiodinium natans** | Unnamed | 94% | 64.16 | CAE7360486.1 |
| *Symbiodinium necroappetens* | carB | 92% | 51.21 | CAE7932356.1 |
| *Cladocopium sp. C3** | HSP70 | 49% | 61.59 | ABA28988.1 |
| TRANSLATED RNA | | | | |
| *Symbiodinium sp. CCMP2430* | unnamed | 76% | 96.74 | HBTH01040005.1 |
| *Crypthecodinium cohnii* | BiP | 74% | 84.01 | AF421538.2 |
| *Breviolum minutum SSB01* | Unnamed | 72% | 81.56 | GICE01029930.1 |
| *Cladocopium goreaui* | Unnamed | 68% | 82.90 | ICPI01002597.1 |
| *Durusdinium trenchii* | Unnamed | 39% | 86.42 | ICPJ01013860.1 |
| *Symbiodinium muscatinei* | Unnamed | 62% | 78.81 | GFDR03033717.1 |
| **DNA** | | | | |
| *Symbiodinium kawagutii* | Unnamed | 31% | 68.08 | KC950716.1 |

The correct SBiP1 sequence (Acc. No. OP429585) was used for a BLAST analysis against all annotated GenBank sequences.

*Cytosolic location.

## The light stimulated SBiP1 dephosphorylation on Thr in CassKB8 cells is not affected by the inhibition of photosynthesis

We further tested whether the light induced SBiP1 dephosphorylation was dependent on photosynthesis by adding DCMU, an inhibitor of PSII, or vehicle alone (EtOH) 2 h prior to the onset of light (Fig 7). We observed the fully Thr phosphorylated SBiP1 band present in the cells after 12 h of darkness in either condition: in the presence of previously added (2 h before the light stimulus) DCMU (Fig 7A, right upper panel (SBiP1-p), Dark, 12h; Fig 7B, black bar, DCMU); or vehicle alone (Fig 7A, left upper panel (SBiP1-p), Dark, 12 h; Fig 7B, black bar, EtOH). Then, the typical light induced SBiP1 dephosphorylation on Thr was observed after exposure for 30 and 60 min in the presence of either DCMU (Fig 7A, right upper panel (SBiP1-p), Light, 30 and 60 min, respectively; Fig 7B, cross-hatched white bars, DCMU, respectively), or when only vehicle was added (Fig 7A, left upper panel (SBiP1-p), Light, 30 and 60 min, respectively; Fig 7B, white bars, Control, respectively). The addition of DCMU showed a clear effect on the photochemical efficiency which remained at marginal levels 30 and 60 min after the onset of the light phase (S6 Fig., dotted line), whereas the untreated cells showed a clear increase in this parameter after the same time intervals when light ensued (S6 Fig., solid line). These data indicated that the light stimulated SBiP1 dephosphorylation on Thr in dark adapted CassKB8 cells occurs independent of photosynthetic related processes, at least within the assayed times.

## Discussion

### Correct annotation, sequence features and ubiquitous presence of SBiP1 in Symbiodiniaceae

We previously reported the identification of a BiP protein from *Symbiodinium microadriaticum* CassKB8 [17, 20]. Our initial observations revealed that the GenBank annotated

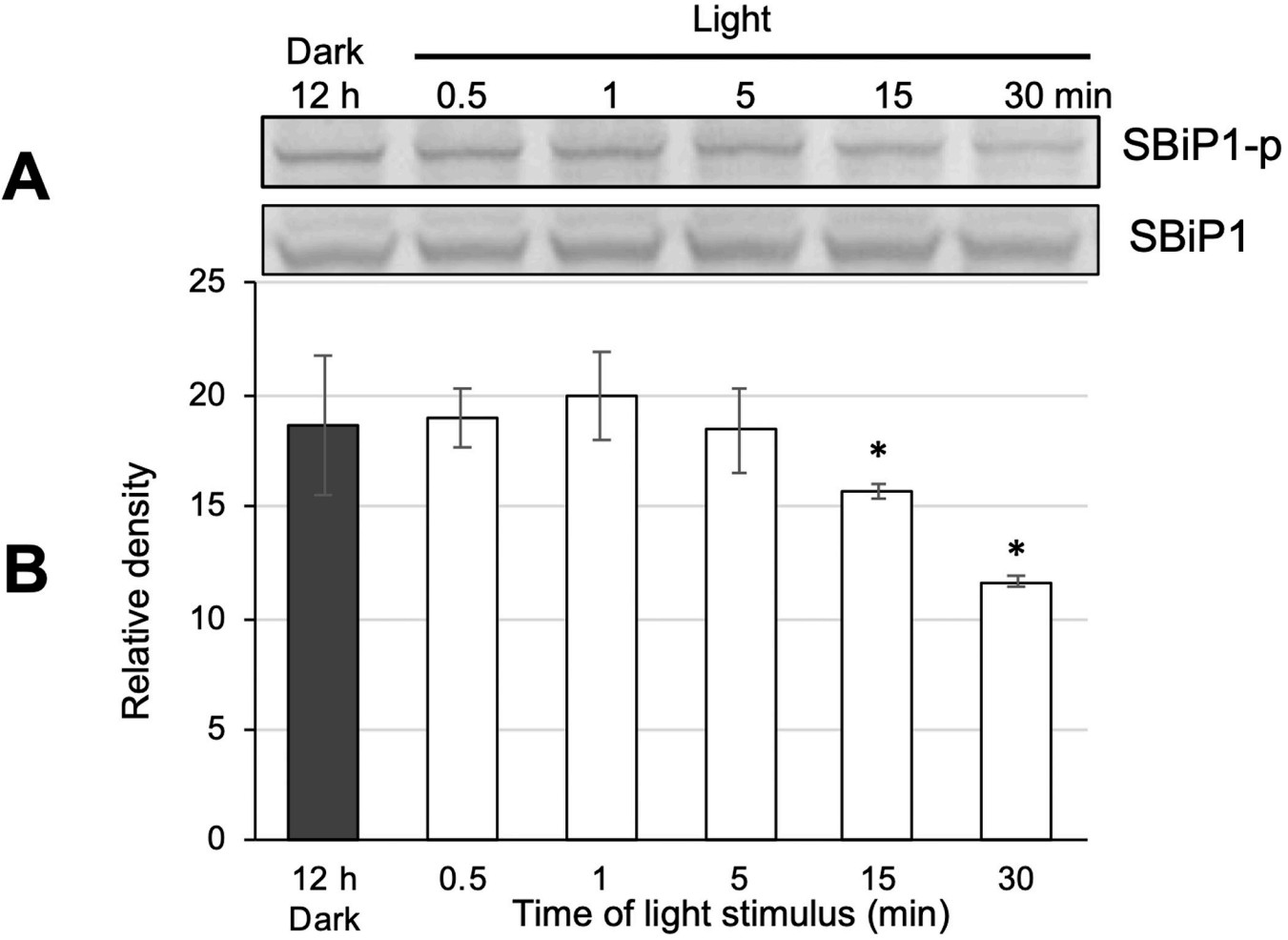

**Fig 4. Time-course of Thr dephosphorylation of SBiP1 after the light stimulus.** A. Upper lanes: western blot with anti-pThr antibodies (lanes SBiP1-p) of total protein extracts from CassKB8 cells adapted to darkness for 12 h (Dark) or after light stimulation for 0.5, 1, 5, 15, or 30 min (lanes 0.5–30 min); lower lanes: western blot with anti-SBiP1 antibodies (lanes SBiP1) of equivalently loaded and treated total protein extracts from CassKB8 cells. B. Densitometric analysis of the SBiP1 Thr phosphorylation level quantitated and normalized to the total SBiP1 levels from three biological replicates of each equivalent treatment.

homologous sequence for *S. microadriaticum* from *Stylophora pistillata* yielded a >200 kDa translated protein product; however, our observations of the molecular weight of 75 kDa of the same protein on SDS-PAGE gels, in addition to multiple alignment analysis of the sequence allowed us to find a misannotated stop codon and determine the correct coding sequence of 2,067 nucleotides and a 689 amino acid protein based on its apparent molecular weight [20]. Nonetheless, further analysis after sequencing of the PCR amplified ORF, revealed that the correct coding sequence actually harbors 1,965 nucleotides and translates to a 655 amino acid product (Fig 1; S1 and S5 Figs). To prevent future confusion due to the misannotated sequence, we submitted it to the GenBank (Acc. No. OP429595). We also obtained the genomic sequence and upon comparison with the CassKB8 database [25], we observed slight changes between them; namely, one non- and one synonymous substitution (S5 Fig). Even though the CassKB8 cells were obtained from the same source for this work and for the genome sequencing [25], these slight changes may have occurred through time since we have kept sub-culturing the cells for longer than twenty years in our collection.

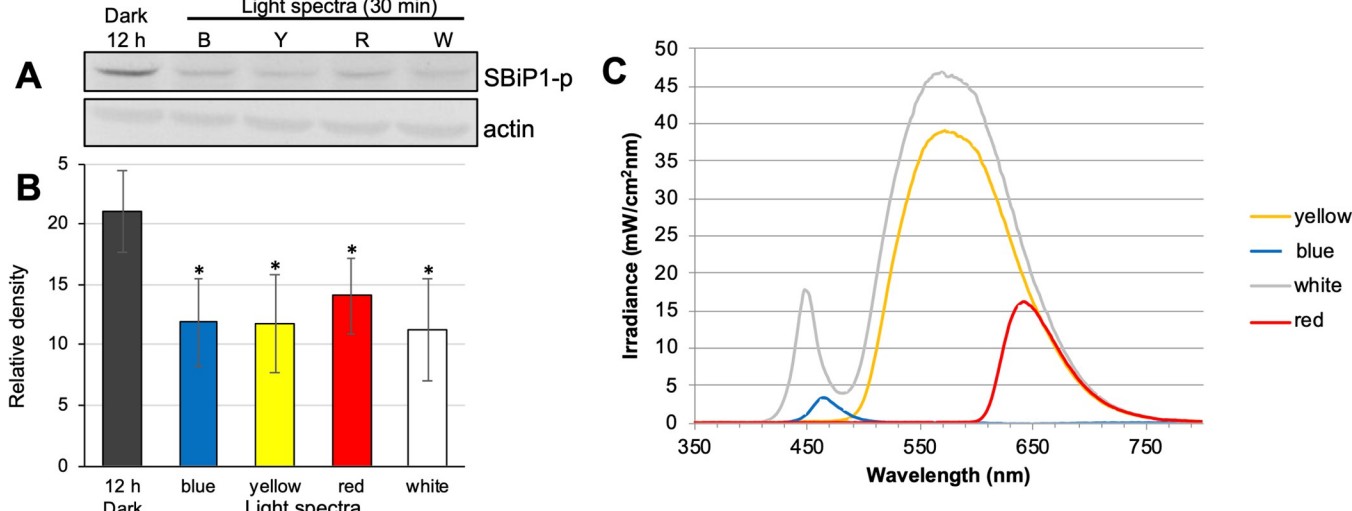

**Fig 5. SBiP1 Thr dephosphorylation levels after stimulation with various light spectra.** A. Upper lanes: western blot with anti-pThr antibodies (lanes SBiP1-p) of total protein extracts from CassKB8 cells adapted to darkness for 12 h (Dark) or after light stimulation with ~ 100 μmole photon m$^{-2}$ s$^{-1}$ of blue, yellow, red, or regular white (lanes B, Y, R, W, respectively) light; lower lanes: western blot with anti-SBiP1 antibodies (lanes SBiP1) of equivalently loaded and treated total protein extracts from CassKB8 cells. B. Densitometric analysis of the SBiP1 Thr phosphorylation level quantitated and normalized to the total actin levels from three biological replicates of each equivalent treatment. C. Graph showing the wavelength ranges of the different lights used for stimulation of the dark-adapted CassKB8 cells, each colored according to their visible spectra, except for the white which is shown in grey.

At the amino acid level, the main molecular features included the typical domains of a protein from the BiP/HSP70 family that contained a C-terminal signal peptide and the N-terminal ER-localization HDEL sequence, indicating its ER residence. The same sequence was found in the reported Symbiodiniaceae BiP homologs from GenBank. This sequence shows variation among kingdoms and even organisms from the same kingdom since Lys-Asp-Glu-Leu (KDEL) predominates in vertebrates, *Drosophila melanogaster* and *Caenorhabditis elegans*, whereas HDEL in most plants and *Saccharomyces cerevisiae* [42]; however, both forms are also present in vertebrates and some plants [42]; thus, the HDEL ER-localization sequence appears to be the rule for Symbiodiniaceae. On the other hand, among the several predicted phosphorylation sites, various Ser and Thr sites but only a Tyr residue at position 309, were detected. However, this Tyr residue appears substituted by Phe in most plants including *Arabidopsis thaliana* and *Solanum tuberosum* [43], *Oryza sativa* and *Zea mays* [44], and *Triticum aestivum* [45]. Interestingly, in BiP from parasite apicomplexans the same residue is substituted by a Phe in *Plasmodium falciparum* [46], or by a Phe or a Thr in *Trypanosoma cruzi* isovariants [47]. Conversely, in *C. reinhardtii* this Tyr is conserved at position 313 (UniProt Acc. No. A8I7T8). In this freshwater dinoflagellate microalga, the main target for phosphorylation has been experimentally demonstrated to be the highly conserved Thr 520 [15]. This Thr was found at position 513 in CassKB8 (Fig 1B) and exposed on the surface of the 3D structure readily available for phosphorylation (Fig 2). This residue was also conserved at equivalent positions in all retrieved Symbiodiniaceae sequences from the GenBank. At least four other Thr residues at positions 87, 312, 319, and 529 were surface exposed on the SBiP1 3D structure; from these, Thr 87, 319, and 529 were also highly conserved and present in mammalian BiP [16]. Taken together, these data along with the observation of at least two Thr phosphorylated isovariants of SBiP1 in CassKB8 [20], suggest that the main phosphorylation events that regulate the light induced chaperone activity occur on various Thr sites including Thr 513.

ADP-ribosylation is another important post-translational modification of BiP chaperones as it has been shown to occur on mouse BiP as an activation switch responsive to the cellular

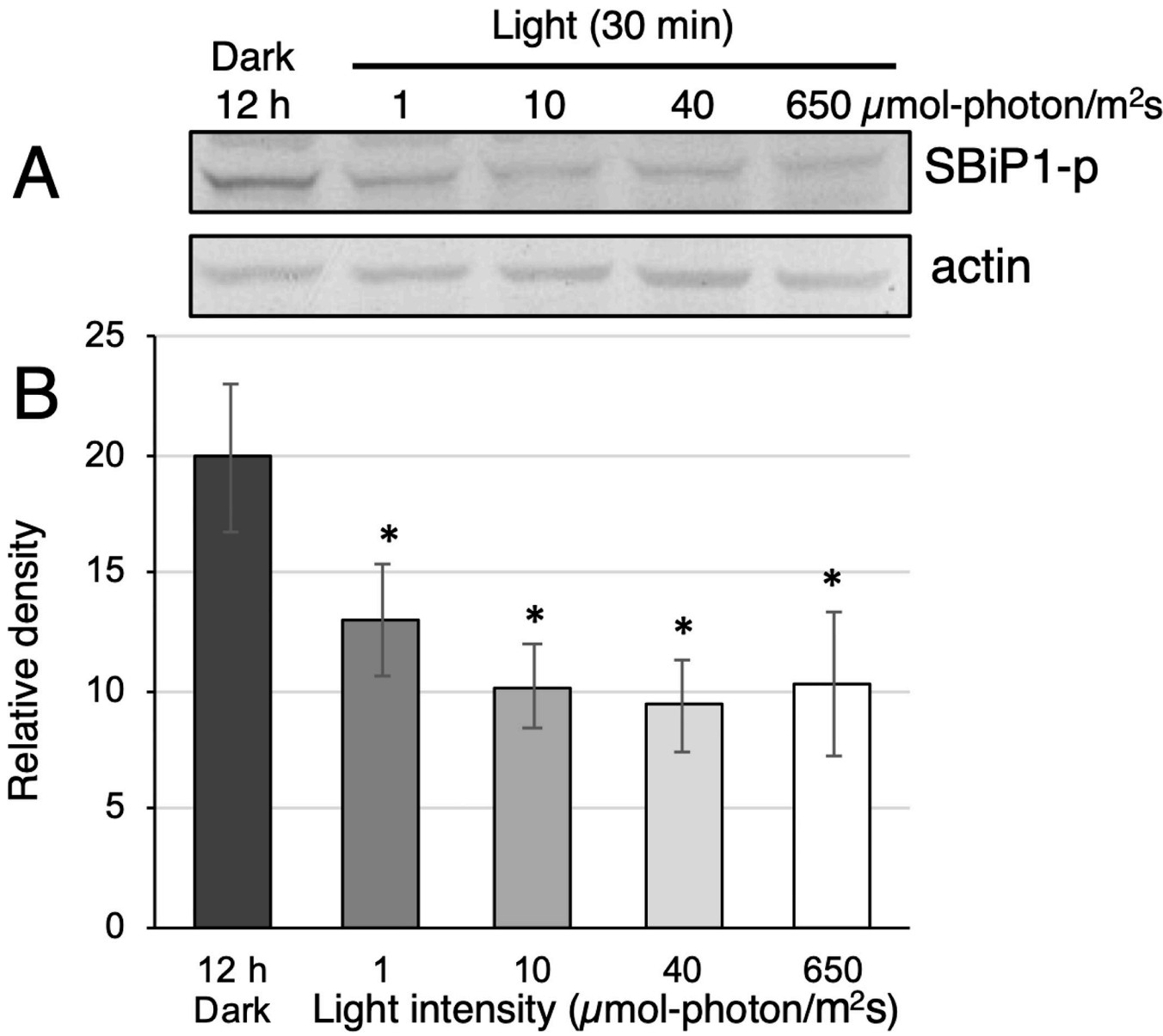

**Fig 6. Thr dephosphorylation levels after stimulation at various light intensities.** A. Upper lanes: western blot with anti-pThr antibodies (lanes SBiP1-p) of total protein extracts from CassKB8 cells adapted to darkness for 12 h (Dark) or after light stimulation at 1, 10, 40, or 650 µmole photon $m^{-2}$ $s^{-1}$ (lanes 1, 10, 40 and 650, respectively); lower lanes: western blot with anti-actin antibodies (lanes actin) of equivalently loaded and treated total protein extracts from CassKB8 cells. B. Densitometric analysis of the SBiP1 Thr phosphorylation level quantitated and normalized to the total actin levels from three biological replicates of each equivalent treatment.

nutritional status [16, 40]; i.e., ADP-ribosylated BiP levels rise in the inactive pancreas of fasted mice whereas the modification decays in well fed animals [40]. This would then correlate with ADP-ribosylation for the correct protein folding at low levels of *de novo* synthesis, whereas deribosylation occurring during increased protein synthesis prevents the aggregation of unfolded proteins [14, 40]. This modification adds another level of regulation complexity on the chaperone in response to nutritional cues since inactivation/activation of *C. reinhardtii* BiP by phosphorylation/dephosphorylation on Thr 520 (513 in SBiP1), respectively, has been proposed to occur also as an activation/inactivation switch responsive to the nutritional

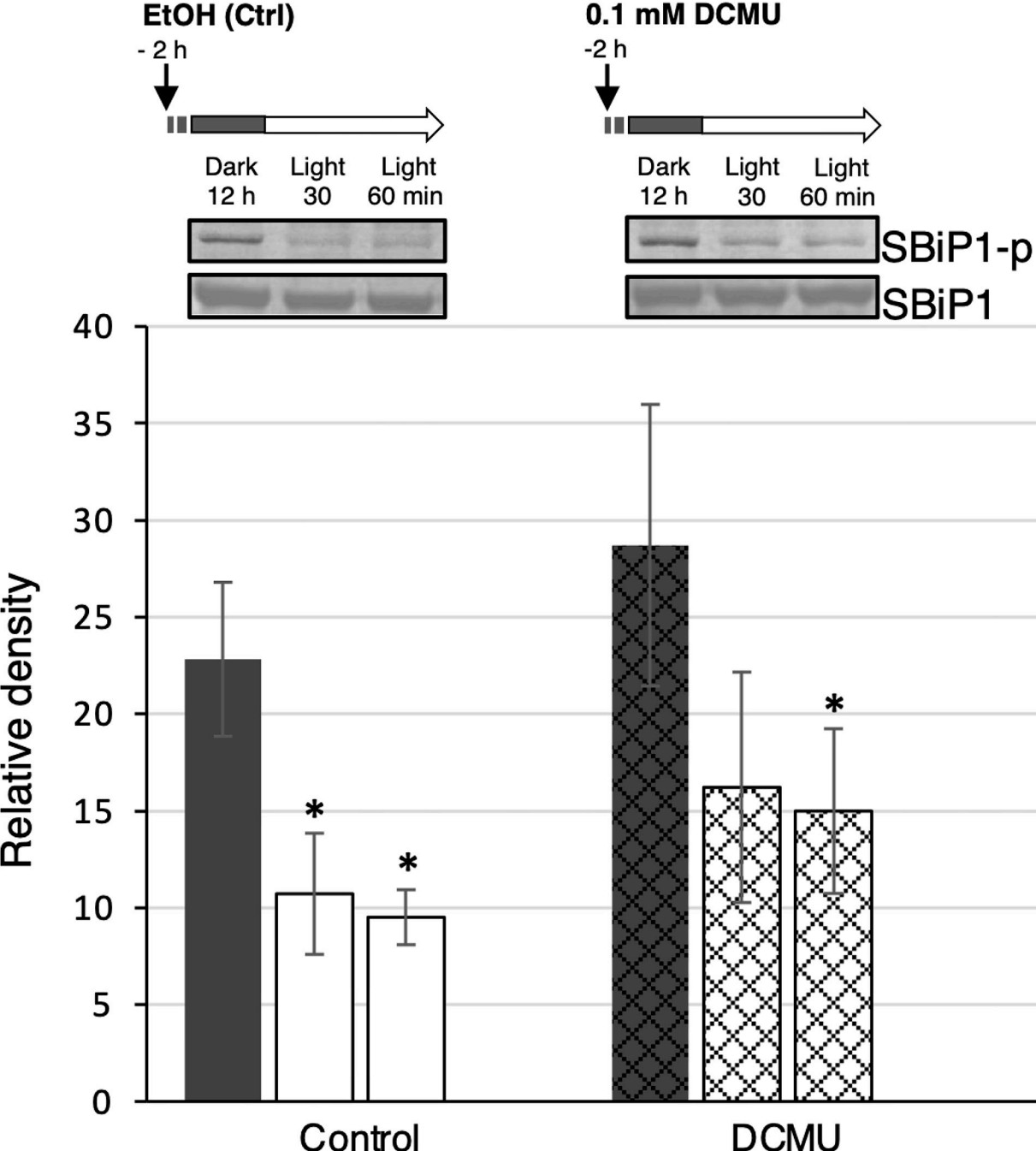

**Fig 7. Effect of 3-(3,4-dichlorophenyl)-1,1-dimethylurea on the Thr phosphorylation behavior of SBiP1 during the dark-to-light transition.** A. Upper panels: representative western blots showing the level of Thr phosphorylation of SBiP1 (SBiP1-p) from dark-adapted CassKB8 cells incubated for 2 h before the light stimulation with 0.01 mM 3-(3,4-dichlorophenyl)-1,1-dimethylurea (DCMU), or vehicle (EtOH) only, and after 30 and 60 min of light. SBiP1 is highly Thr-phosphorylated after 12 h of continuous darkness regardless of the treatments (SBiP1-p; Dark, 12 h). SBiP1 Thr phosphorylation levels from dark adapted cells decrease significantly after a 30- or 60-min light stimulus in the presence of either vehicle (SBiP1-p, left panel; lanes 30, 60 min, respectively) or DCMU (SBiP1-p, middle panel; lanes 30, 60 min, respectively). Lower panels: western blots of the same treatments and time points analyzed with anti-SBiP1 antibodies (SBiP1) as internal protein loading controls and used for normalization of the densitometric analysis, showing constant levels of the SBiP1 protein throughout the treatments. B. Densitometric analysis of three biological replicates for each treatment quantitatively showing the levels of SBiP1 Thr phosphorylation under darkness for all three treatments (black bars), or after 30- and 60-min light stimulation in the presence of vehicle (white bars), or DCMU (cross-hatched bars).

cellular status [15]. Mouse BiP has two conserved Arg 471 and 493 residues (Uniprot Acc. No. P20029) where this modification takes place [40]. Although these modifications have not been experimentally shown to occur on SBiP1 and these sites were not revealed by a program for prediction of ADP-ribosylation sites [33], these conserved Arg residues were also present at positions 465 and 487 in CassKB8 SBiP1 and their equivalent positions were detected in all Symbiodiniaceae sequences reported so far in the GenBank. These residues were also surface exposed on the 3D structure (Fig 2) and thus, readily available to modifying enzymes. Finally, the protein was named SBiP1 as it was previously found in three Symbiodiniaceae species experimentally analyzed [17, 20], and our latest screen detected it in all reported sequences from the GenBank (Table 1). Thus, our current knowledge further indicates that SBiP1 is ubiquitous in Symbiodiniaceae.

### SBiP1 dephosphorylation is highly sensitive to white light and occurs under broad visible light spectra

BiP chaperones are known to undergo several post-translational modifications including phosphorylation [16]. Furthermore, it has been reported that they switch from inactive to active states via phosphorylation/dephosphorylation [15]. SBiP1 displays a light responsive phosphorylation behavior; that is, highly phosphorylated on Thr under darkness and dephosphorylation upon light exposure [17, 20], which is consistent with the notion of this activity switch. Since Symbiodiniaceae can live free in the water column or in symbiosis with other marine organisms and in either case light is the trigger for many cellular responses, they must be capable of detecting variations in the light stimuli occurring throughout the day, as well as their complete absence. Furthermore, we expected that blue light would be particularly stimulating with respect to other light spectra since it is the most penetrating form of light into the water column. Surprisingly, the light stimulated Thr dephosphorylation of SBiP1 occurred under either red, blue, or yellow light spectral wavelengths (Fig 5). These results rule out that a canonical cryptochrome- or phytochrome-related light receptor is located upstream the phosphatase that dephosphorylates SBiP1. Broad wavelength perception of blue, green, orange, red, and far-red spectra has been reported for phytochrome-related light receptors from algae [48], and similar broad wavelength receptors could exist in Symbiodiniaceae. Rockwell and collaborators [48], proposed that in places near the surface the red-light perception might be optimized whereas at depth, the red-light intensity is attenuated, and the perception of blue light would be performed by the blue-shifted dark states of the corresponding receptors. This is consistent with the preferential association of Symbiodiniaceae with shallow water organisms such as those in the reef environment. Furthermore, we found a regulatory element for light signaling mediated by phytochrome by *in silico* analysis of the *SBiP1* genomic sequence (Fig 3B). Finally, another requirement of Symbiodiniaceae as underwater organisms that are deep within the host tissue and subjected to light attenuation plus environmental wavelength changes in the water column, would be a high sensitivity to light perception. In this regard, we found that the Thr dephosphorylation response of SBiP1 in CassKB8 was highly sensitive to light since an intensity of only 1 μmole photon m$^{-2}$ s$^{-1}$ of white light was sufficient to trigger the light-stimulated activation of the chaperone (Fig 6). Thus, our data suggest that a high-sensitivity phytochrome receptor of broad range perception could be involved in the initial step of this signal-transduction cascade occurring in Symbiodiniaceae. It is important to emphasize that our results were obtained from cultured Symbiodiniaceae, and future experiments will be required to determine whether a similar SBiP1 behavior is observed *in hospite* in *C. xamachana* endosymbionts. Furthermore, similar experiments in different cnidarian-dinoflagellate symbiotic associations are needed to understand how evolutionary adaptations occurred to

optimize phosphorylation and other post-translational modifications responding to light under the influence of water column depth and other light attenuation parameters.

## Photosynthetic function is not a requirement for the light stimulated dephosphorylation of SBiP1

Disruptors of the photosynthesis machinery in photosynthetic organisms usually target light stimulated pathways. However, this is not always the case since there are examples of other light-mediated mechanisms that occur independent of photosynthesis. In plants for example, light stimulated expression of the *WUSCHEL* gene, a regulator of stem cell differentiation, occurs independently of the photosynthetic process [49]. DCMU is known to compete with plastoquinone for its binding site on PSII [50] and thus, able to disrupt photosynthesis in photosynthetic organisms including Symbiodiniaceae (S6 Fig). Interestingly, the light stimulated Thr dephosphorylation of SBiP1 still occurred in the presence of DCMU (Fig 7A and 7B), indicating that photosynthesis was not necessary for the chaperone activation by the light stimulus. It is also possible that the sudden burst of light when the light phase of the growth cycle ensues, produces an initial oxidative stress that triggers the chaperone activation. For example, it is known that in plants, both light and heat stress result in oxidative stress that induces reactive oxygen species (ROS), and that HSP70 is necessary for protection against such stress [51].

A detailed analysis of the SBiP1 molecular features has revealed important information that can shed light on the possible mechanism of the light-mediated chaperone activation/inactivation by post-translational modification. Initially, the chaperone is fully phosphorylated during the cell dark cycle. Upon entry to the light cycle, a broad-range highly sensitive phytochrome-like receptor triggers the signal-transduction cascade mediated by yet unknown molecules. The cascade involves the activation of an unknown phosphatase that dephosphorylates SBiP1, likely on the highly conserved Thr 513, and would activate the chaperone rendering it functional for the *de novo* protein synthesis brought about by the onset of photosynthesis. This BiP chaperone activation has been demonstrated to occur specifically by dephosphorylation on the equivalent Thr 520 from *C. reinhardtii* in response to stimulation of protein synthesis [15]. Although in that case, the *Cr*BiP phosphorylation responses modulated by light were not determined, it will be interesting to determine the light modulated SBiP1 phosphorylation in response to nutritional cues. ADP-ribosylation could be an additional activation switch; however, whether this modification is also light-modulated remains to be determined. It will be fundamental to dissect the putative receptor, signal-transduction effectors, and kinase(s) and phosphatase(s) involved in this light-activated signaling cascade.

## Supporting information

**S1 Table. Oligonucleotides used to amplify the *SBiP1* ORF and promoter, as well as for sequencing.**
(DOCX)

**S1 Fig. Correct nucleotide and amino acid sequence of SBiP1 after careful analylis of all reported homologous sequences from the GenBank.** The sequence was annotated with accession number OP429595.
(PDF)

**S2 Fig. SignalP 5.0 results of the SBiP1 amino acid sequence.** According to the software, there is a 0.998 likelihood for a cleavage site between Arg18 and Lys19 (green dashed line) for SPase I (Sec/SPI) (red continuous line).
(PDF)

**S3 Fig. Multiple sequence alignment of SBiP1 of the different regions with amino acids potentially relevant to SBiP1 function.** All highlighted in black boxes: two predicted phosphorylation sites, a Tyr residue at position 309, and an experimentally demonstrated highly conserved Thr residue (T513); in addition, Arg residues at position 465 and 487 represent putative ADP-ribosylation sites. Positions are numbered according to the *Symbiodinium microadriaticum* CassKB8 SBiP1 sequence.
(PDF)

**S4 Fig. Superimposition of BiP1 and *Homo sapiens* BiP structures.** The SBiP1 3D structure obtained by AlphaFold2 (red) was superimposed on the 3D structure of *H. sapiens* BiP (cyan; PDB: 5e84). A close structural homology between both 3D structures is observed.
(PDF)

**S5 Fig. Intron/exon organization of the *SBiP1* gene.** On top, schematic representation of intron/exon organization of the *SBiP1* gene. On the bottom, the coding sequence of the *SBiP1* gene (cDNA) compared to the reported (*25*) genomic sequence (gDNA). Nucleotides are numbered according to the cDNA sequence; the start and stop codons are shown in boldface, and introns are shortened within the >>>....>>> characters. The amino acid sequence is displayed in one-letter code. We found one substitution (A130C, shaded in orange) that changed the His44 for an Asn, and one synonymous substitution (A780G, Gln260Gln, shaded in yellow).
(PDF)

**S6 Fig. Photochemical efficiency of CassKB8 Photosystem II in the presence of DCMU.** CassKB8 PSII photochemical efficiency with (dotted line) or without (continuous) DCMU from 2 h before the transition from dark to light and up to the first hour after the light onset (schematically depicted with the black/white lower bar. The cells without DCMU show the expected behavior of increase in the photochemical conversion with light whereas those with DCMU show constant low values throughout. Measurements show the average of three biological replicates.
(PDF)

## Acknowledgments

We are grateful for the technical help of M.I. Miguel Ángel Gómez-Reali, M.O. Edgar Escalante-Mancera, M.T.I.A. Gustavo Villarreal-Brito, Dr. Edén Magaña-Gallegos and M.C. Laura Celis-Gutiérrez.

## Author Contributions

**Conceptualization:** Marco A. Villanueva.

**Formal analysis:** Raúl Eduardo Castillo-Medina, Tania Islas-Flores, Estefanía Morales-Ruiz.

**Funding acquisition:** Marco A. Villanueva.

**Investigation:** Raúl Eduardo Castillo-Medina, Tania Islas-Flores, Estefanía Morales-Ruiz.

**Methodology:** Raúl Eduardo Castillo-Medina.

**Supervision:** Marco A. Villanueva.

**Validation:** Raúl Eduardo Castillo-Medina.

**Writing – original draft:** Marco A. Villanueva.

**Writing – review & editing:** Raúl Eduardo Castillo-Medina, Tania Islas-Flores, Estefanía Morales-Ruiz, Marco A. Villanueva.

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
