## [Decision Letter · Decision Letter 0]

19 Sep 2023

PONE-D-23-23053Biochemical and molecular characterization of the SBiP1 chaperone from *Symbiodinium microadriaticum* CassKB8 and light parameters that modulate its phosphorylationPLOS ONE

Dear Dr. Villanueva,

Thank you for submitting your manuscript to PLOS ONE. After careful consideration, we feel that it has merit but does not fully meet PLOS ONE’s publication criteria as it currently stands. Therefore, we invite you to submit a revised version of the manuscript that addresses the points raised during the review process.

We look forward to receiving your revised manuscript.

Kind regards,

Cheorl-Ho Kim, Ph.D.

Academic Editor

PLOS ONE

Journal Requirements:

Additional Editor Comments:

Dear Dr Villanueva,

Ref: Biochemical and molecular characterization of the SBiP1 chaperone from Symbiodinium microadriaticum CassKB8 and light parameters that modulate its phosphorylation.

I have completed the evaluation of your manuscript. As our reviewers have also recommended, your study is valuable for publication.

Personally, your present study should be documented after a minor revision as simple points.

I would appreciate you for your submission. In future, I also look forward to receiving your studies.

Thank you

Sincerely

Cheorl-Ho Kim PhD Professor

SKKU

Editor

Reviewers' comments:

Reviewer's Responses to Questions

**Comments to the Author**

1. Is the manuscript technically sound, and do the data support the conclusions?

Reviewer #1: Yes

Reviewer #2: Yes

2. Has the statistical analysis been performed appropriately and rigorously? 

Reviewer #1: Yes

Reviewer #2: Yes

3. Have the authors made all data underlying the findings in their manuscript fully available?

Reviewer #1: Yes

Reviewer #2: Yes

4. Is the manuscript presented in an intelligible fashion and written in standard English?

Reviewer #1: Yes

Reviewer #2: Yes

5. Review Comments to the Author

Reviewer #1: The manuscript described a characterization study of a Hsp70 family chaperone SBiP1 from a dinoflagellate. The authors first corrected the sequence and annotation of the protein and then analyzed the domains and motifs in the protein using bioinformatics. The authors found that SBiP1 showed dephosphorylation upon the light stimulation, despite the wavelength. They further showed evidence that the light-stimulated SBiP1 dephosphorylation does not depend on photosynthesis. The results are well presented and the conclusion is well supported by the data.

Here are some suggestions to improve the manuscript.

1. The first section in Results: the reason for the sequence correction should be explained before reporting the correct ORF.

2. Line 267: The result of SignalP signal peptide prediction should be given in detail (the resulting figure of SignalP, the values of possibility, or other predicted parameters).

3. Line 138: Describe the sequence analyses in a new section.

4. Line 293: a useless parenthesis.

5. Line 469-471: Asn is not a basic amino acid. There is no evidence to support the sentence, and it should be removed.

Reviewer #2: Castillo-Medina et al have undertaken classical proteomic approaches and combined with physiological experiments to investigate the role of SBiP1 as a chaperone. During this investigation, they uncovered the sequences level sites that undergo post-translational modifications and quantified proteomically such events. This type of work is relatively rare in the dinoflagellate community and this is commendable that authors have undertaken it.

There are areas such would have enhanced their work such as undertaking a phylogenetic confirmation of this protein or reanalyzing RNAseq data from public sources to support the role of SBiP1. There are minor sections that need rephrasing to avoid confusion to readers and places to provide clarity. Post-transcriptional and post-translation control in dinoflagellates are poorly known and investigated and this work does attempt to investigate one of these.

One area which is difficult to address is the role of post-transcriptional and post-translation control of certain transcripts & proteins. I wonder if SBiP1 suffer from such control. A potential idea to explore later will be to look at RNAseq data and expression of SBiP1, especially under different treatments.

Abstract:

Line 29-30: This sentence confused me. Please rephrase it.

Introduction:

Line 75-77: 869 sequences protein kinases sequences are described in González-Pech et al (2017) [11] only while Supplementary Table 12 in Liu et al (2018) [12] gives a more comprehensive representation of these domains in Symbiodiniaceae. It might be helpful to express the abundance of kinase domains as a range or percentage within Symbiodiniaceae. With more Symbiodiniaceae genomes, this sentence provides larger context.

Line 100-102: It might be helpful to readers if this sentence can be rephrased to clarify that dephosphorylation of BiP correlates with chaperone activation (Díaz-Troya S et al [15]) or authors can justify why they termed it as "presumptive".

Methods:

Line 167-168: Please describe briefly how the substrate binding domain and the ADP-ribosylation sites were determined.

Line 213-214: How was the light intensity controlled? Was it in automated option in an incubator?

Line 278-279: Please show the conservation plot of the phosphorylation residues as a supplementary figure; it can be confirmed for Thr 513 as shown in figure 5 in [15] but there is no plot for Tyr 309. The same applies for Arg 465 and 487. A simple alignment should be enough.

Line 291-293: In Fig S2 both structures are difficult to differentiate due to the color. Can the authors show one of the structures in a different color to ease visualisation.

Line 352-265: It is difficult to confirm or ascertain what authors are claiming in terms of dephosphorlyation level from the blot. I urge the authors to add a statement to tone down this claim. I am glad that Line 356-369 does support this claim statistically.

Discussion:

Line 469: Typo "synonimous"

Line 522: Typo "dephsophorylation"

Line 587-589: I would encourage the authors to give a broader perspective to their discussion especially to adaptation to Symbiodiniaceae in different symbiotic relationship. In this case the strain was obtained from jellyfish. It should be possible to look at Symbiodiniaceae RNAseq data publicly available to support these observations. There is a brief mention to shall water Symbiodiniaceae ( Line 544-545). The role of such post-translational events in key proteins must have been an evolutionary advantage for symbiosis especially with corals.

6. PLOS authors have the option to publish the peer review history of their article (what does this mean?). If published, this will include your full peer review and any attached files.

Reviewer #1: **Yes: **Yingang Feng

Reviewer #2: No

---

## [Author Response · Author response to Decision Letter 0]

4 Oct 2023

Reviewer No. 1

1. The first section in Results: the reason for the sequence correction should be explained before reporting the correct ORF.

Response: We have modified the text to mention the reason for sequence correction before reporting the correct ORF (lines 272-277).

2. Line 267: The result of SignalP signal peptide prediction should be given in detail (the resulting figure of SignalP, the values of possibility, or other predicted parameters).

Response: A supplementary figure S2 with the results has been added and a brief description mentioned in the methods (lines 169-171 and 279).

3. Line 138: Describe the sequence analyses in a new section.

Response: The sequence analyses have been separated and are described in a new section (line 134 and 164).

4. Line 293: a useless parenthesis.

Response: The parenthesis has been deleted (line 305).

5. Line 469-471: Asn is not a basic amino acid. There is no evidence to support the sentence, and it should be removed.

Response: The line has been deleted (line 482).

Reviewer No. 2

Query: Line 29-30: This sentence confused me. Please rephrase it.

Response: It has been rewritten for clarity (lines 24-25).

Query: Line 75-77: 869 sequences protein kinases sequences are described in González-Pech et al (2017) [11] only while Supplementary Table 12 in Liu et al (2018) [12] gives a more comprehensive representation of these domains in Symbiodiniaceae. It might be helpful to express the abundance of kinase domains as a range or percentage within Symbiodiniaceae. With more Symbiodiniaceae genomes, this sentence provides larger context.

Response: The kinase domains have been expressed as a range and “sequences” has been changed to “families” as correctly reported (lines 71-72).

Query: Line 100-102: It might be helpful to readers if this sentence can be rephrased to clarify that dephosphorylation of BiP correlates with chaperone activation (Díaz-Troya S et al [15]) or authors can justify why they termed it as "presumptive".

Response: The term ”presumptive” has been deleted, the words “implying its activation” was added and the corresponding references (Díaz-Troya et al. 2011 [15] and Nitika et al. 2020 [16] were already provided in the sentence (lines 97-98).

Query: Line 167-168: Please describe briefly how the substrate binding domain and the ADP-ribosylation sites were determined.

Response: A brief description was included (lines 173-174 and 176-177).

Query: Line 213-214: How was the light intensity controlled? Was it in automated option in an incubator?

Response: A sentence with this information has been added in the corresponding Methods section (lines 226-228).

Query: Line 278-279: Please show the conservation plot of the phosphorylation residues as a supplementary figure; it can be confirmed for Thr 513 as shown in figure 5 in [15] but there is no plot for Tyr 309. The same applies for Arg 465 and 487. A simple alignment should be enough.

Response: A supplementary figure showing the alignment and a slight change in the text was made (line 290; S3 Fig. mentioned in lines 291 and 294).

Query: Line 291-293: In Fig S2 both structures are difficult to differentiate due to the color. Can the authors show one of the structures in a different color to ease visualisation.

Response: A figure with contrasting colors now S4 Fig. has replaced the former S2 (mentioned in line 304).

Query: Line 352-265: It is difficult to confirm or ascertain what authors are claiming in terms of dephosphorylation level from the blot. I urge the authors to add a statement to tone down this claim. I am glad that Line 356-369 does support this claim statistically.

Response: The statement has been toned down to say “and was still observed at apparently higher levels after 30 min of light (line 368).

Query: Line 469: Typo "synonimous"

Response: The sentence was deleted at the request of Reviewer 1 (line 482) and the same typo was corrected from line 479.

Query: Line 522: Typo "dephsophorylation"

Response: Typo was corrected to “dephosphorylation” (line 533).

Query: Line 587-589: I would encourage the authors to give a broader perspective to their discussion especially to adaptation to Symbiodiniaceae in different symbiotic relationship. In this case the strain was obtained from jellyfish. It should be possible to look at Symbiodiniaceae RNAseq data publicly available to support these observations. There is a brief mention to shall water Symbiodiniaceae (Line 544-545). The role of such post-translational events in key proteins must have been an evolutionary advantage for symbiosis especially with corals.

Response: In this regard, we chose not to enter into too much speculation on the subject mentioned by the reviewer since the experiments from our work were carried out in cultured Symbiodiniaceae. Instead, we emphasized this fact and suggested that further experimentation is required on endosymbionts from diverse holobiont systems including C. xamachana to be able to understand evolutionary adaptations. (lines 566-572).

---

## [Editor Report · Decision Letter 1]

10 Oct 2023

Biochemical and molecular characterization of the SBiP1 chaperone from *Symbiodinium microadriaticum* CassKB8 and light parameters that modulate its phosphorylation

PONE-D-23-23053R1

Dear Dr. Villanueva,

We’re pleased to inform you that your manuscript has been judged scientifically suitable for publication and will be formally accepted for publication once it meets all outstanding technical requirements.

Kind regards,

Cheorl-Ho Kim, Ph.D.

Academic Editor

PLOS ONE

Additional Editor Comments (optional):

Dear Marco

I should thank you for your revision and original submission to PLOS ONE.

I think that your revision is appropriately done and can be accepted for publication.

I recognize that no more circulatiom to further evaluate the revision through second round review process is needed.

Thus, I would like to accept your revision.

I look forward to reading your study upon publication.

Thanks a lot

Cheorl-Ho Kim PhD Professor

Editor
---

## [Editor Report · Acceptance letter]

13 Oct 2023

PONE-D-23-23053R1 

Biochemical and molecular characterization of the SBiP1 chaperone from *Symbiodinium microadriaticum* CassKB8 and light parameters that modulate its phosphorylation 

Dear Dr. Villanueva:

I'm pleased to inform you that your manuscript has been deemed suitable for publication in PLOS ONE. Congratulations! Your manuscript is now with our production department. 

Kind regards, 

on behalf of

Professor Cheorl-Ho Kim 

Academic Editor

PLOS ONE